# Virtual Testing Workflows Based on the Function-Oriented System Architecture in SysML: A Case Study in Wind Turbine Systems

**Yizhe Zhang *** , **Julian Roeder** , **Georg Jacobs** , **Joerg Berroth** and **Gregor Hoepfner**

Institute for Machine Elements and Systems Engineering, RWTH Aachen University, 52062 Aachen, Germany
* Correspondence: yizhe.zhang@imse.rwth-aachen.de

**Abstract:** Wind turbines (WT) are complex multidisciplinary systems containing a large number of mechanical, control, and electrical components. Model-Based Systems Engineering (MBSE) provides an approach for cross-discipline development to address the system complexity and focuses on creating and utilizing domain models as the primary means of information exchange. The domain models predict system behaviors and can support system validation through virtual testing at an early stage of system development. However, the further the WT development proceeds, the more system parameters are set, and the more domain models and virtual tests are involved. Therefore, it is necessary to design a framework of virtual testing workflows of WTs to support virtual validation processes as well as to automate those workflows. To achieve this goal, this contribution shows how standardized virtual testing workflows can be designed and linked to hierarchical and functional system architectures modeled in the Systems Modeling Language (SysML). The virtual testing workflows enable to trigger simulations of domain models and handle system parameters participating in the simulations, thus ensuring data consistency. Furthermore, to facilitate modular management and reuse of domain models, the domain models are classified according to model purposes, model fidelities, and system scopes. The virtual testing workflows are structured corresponding to the classification of the domain model, thereby forming a nested framework. To verify the feasibility of the proposed workflows, a virtual testing process of WT components (i.e., bearings) inside the system context with different model purposes and different model fidelities is demonstrated. It is shown that virtual testing workflows are systematically organized so that engineers can easily virtually (re-)validate the systems.

**Keywords:** model-based systems engineering; wind turbine system; virtual testing workflow

## 1. Introduction

The analysis of energy scenarios shows that wind energy will play an increasingly important role in the energy supply system in the future [1]. A wind turbine (WT) is a multidisciplinary system including mechanical components (e.g., drivetrain), electrical components (e.g., generator), control components (e.g., controller), etc., that converts wind energy into electrical energy. These components are closely interconnected with each other to always ensure high-performance conditions for the WT system in different wind conditions [2,3]. Therefore, in the process of multidisciplinary system (such as a WT system) development, engineers are required not only to focus on the design of their system within the specific discipline, but also to consider the dependencies with other disciplines in order to achieve a balanced system solution [4]. However, with the increase in complexity in the later stage of system development, there are more and more implicit dependencies between designs. In the face of the rapid growth in the scale and complexity of multidisciplinary systems, being able to develop systems under ever-faster changing and more individual market requirements is becoming more and more challenging [5]. In this case, when the engineers redesign in order to meet specific requirements or solve the occurrence of a

failure risk, it often results in additional unknown failures caused by the design changes. As an example, developers need to redesign mechanical components (e.g., drivetrain) of WTs to be able to transfer more loads. An appropriate range of rotational speeds ensures the appropriate loading of the redesigned drivetrain. However, the design parameters in the aerodynamics and the control models also need to be considered in order to adjust the rotational speed of the WT rotor [6]. This requires more efficient cooperation between engineers and timely communication of data changes. Therefore, it is becoming more and more important to find a communication method to maintain the seamlessness of data across the various disciplines involved to ensure that a cost-effective WT system is designed to best serve the needs of stakeholders.

In order to ensure the seamlessness of data in the development of multidisciplinary systems such as WTs, system development methods must continue to evolve. Model-based Systems Engineering (MBSE) provides an interdisciplinary collaborative approach for deriving, developing, and testing systems that meet customer expectations in a more efficient way [7,8]. The term Model-based Systems Engineering (MBSE), as introduced by International Council on Systems Engineering (INCOSE), is focused on formalizing the use of models in the field of Systems Engineering to support system requirements, design, analysis, verification, and validation processes throughout a system's lifecycle [9]. The MBSE approach for addressing issues in a multidisciplinary system development process has the potential to guide WT projects to address the above-mentioned challenge. The value of MBSE is based on the fact that all system-related information is configured and linked in a central repository for management. This feature enables domain models, parameters, and related design information from a range of different disciplines to be connected to each other and provides transparency of information over all disciplines. This interconnection can support the automatic propagation of design changes, consistency checks, and failure identification.

The application of MBSE is based on three core pillars which are language, tools, and methods [10]. One language for MBSE is SysML, which provides a general modeling language for its implementation and is commonly used by many developers [11]. The OMG has developed the SysML as an extension of the Unified Modeling Language (UML) to support systems engineering activities by creating and managing models of systems using well-defined blocks, linkages, and other visual constructs. The system model as defined with SysML describes a complete view of the multidisciplinary system under development, including its requirements, behaviors, structures, parameters, and their interactions for the design and validation of systems [12–14]. This system model is used as a centric platform to communicate with various stakeholders [15]. SysML developed by the Object Management Group (OMG) is widely used for abstract modeling of systems engineering problems in the industrial field [16]. A specific SysML modeler, called Cameo Systems Modeler (CSM) [17], used in this work allows the designer to integrate with and execute an external evaluator (e.g., MATLAB), which provides the possibility of interaction between the existing domain simulation models in various simulation tools and the centric system model [18,19]. Domain model-based modeling ensures that engineers can perform high-precision virtual testing in centric system models to support the virtual validation of WT system. The method describes the standardized system development process and how the language should be used to generate the system model. Several used methods for MBSE are summarized in [20]. For instance, the Object-Oriented Systems Engineering Method (OOSEM) is a hybrid approach using object-oriented concepts combined with traditional systems engineering methods. It mirrors the classical "V" [21] lifecycle development model of system design that enables the system engineer to define, specify, and analyze the system among various system views throughout the development process [22]. The object-oriented techniques improve the traceability and extensibility of systems to adapt to evolving technologies and changing needs. In the case of complex mechatronic systems across multi-domain models, the Cyber MagicGrid approach was proposed by NoMagic [23]. Cyber MagicGrid is an extension of MagicGrid [24], which is defined with three layers of abstraction (Problem,

Solution, Implementation) and four pillars (Requirements, Behavior, Structure, Parameters). Cyber MagicGrid allows every team member to share information about design parameters about their tasks during the development process at the detailed layer. In [25], the need for functional system modeling and the necessity of defining hierarchical system functions through SysML elements was observed and the Functional Architecture for system (FAS) method was proposed. Model-based functional architecture description models a system independently of its target technology by transforming the functional elements of modeling information, signals, materials, force, or energy. However, these application methods of MBSE lack a detailed top-down modeling method of systems and there are no traceability links clearly established between system functions and components.

In this work, a practical MBSE method proposed by Jacobs [26] for function-oriented system development is applied. The advantage of this method over other methods is that it allows for transparent traceability by building explicit interdependencies of system architecture between requirements, functions, and the solution layer on the parameter level. In addition, this method implements the integration between system models and domain models from various engineering disciplines, rather than staying at the conceptual design level. An executable system model integrated with the domain model provides the foundation for virtual validation. In the traditional system development process, validation usually is performed at the end of the development stage [27]. However, it is hard to make changes in the later stages of development for complex systems, because this often means that they are unaffordable in terms of time or budget or require more effort than designing from the beginning all over again [28]. Virtual testing provides a way to simulate the physical behavior early in development, which involves simulation of quantitative dynamic behavior of technical system models in a variety of environments and conditions to virtually validate that the system works properly. Virtual testing uses physical behavior models and methods such as finite element analysis (FEA) and multibody simulations (MBS) to create digital domain models of systems, before ever having to create physical prototypes. Some studies have introduced the virtual tests into the above MBSE method to support the system's virtual validation process [26,29].

However, although the failure identification and virtual validation process can be conducted as early as possible, an experienced engineer is still required to manually sequence the simulation's activities in various virtual tests. This leads to two issues described in the following. At first, for complex technical systems (e.g.,: WT systems), a complete virtual test of the system always involves various domain models with different system scopes and model purposes, such as fatigue analysis models of mechanical components or short-circuit failure models of electrical components. The dependencies between system designs increase exponentially among a large number of domain models, it becomes more and more difficult to ensure that every relevant virtual test is conducted. Secondly, as the system is further developed, the fidelity of the simulation model will evolve during the development process. In other words, the further the WT development proceeds, the more product parameters will be set, and the higher the level of detail and fidelity the models have. Incoherent simulation activities often require engineers to manually update system parameters based on virtual testing results to ensure data consistency, which is also inefficient and requires tedious work for complex systems. Therefore, there is still a lack of a solution to build structured virtual testing workflows to support the automated virtual validation process. This work aims to provide a method to design structured workflows in system models based on function-oriented architecture, to ensure the detection of the potential failure risks of the system, improve the reusability of the virtual testing workflows, and the efficiency of the validation process.

In order to achieve the above objective, firstly, this work needs to develop a standardized system architecture for system modeling of WT systems. Formal system modeling bridges the gap between requirements, system functionality, and specific technical solutions, thereby supporting data consistency during virtual testing of systems. The domain models also need to be represented in the system architecture to further describe the solution and be linked to external simulation models to guarantee the accuracy of the test results. Subsequently, external simulation models need to be classified to improve the efficiency and reusability of the model in the virtual testing process. Then, this work needs to propose structured virtual testing workflows in the system architecture. The workflows should be able to sequence the simulation activities of the domain models according to the model classification to support the virtual validation of the redesigned system solutions. The workflows should also provide data for domain models, trigger simulations of domain models in sequence, and keep system data up-to-date. Finally, the workflow design method is implemented in the design and virtual validation process of a high-speed shaft (HSS) bearing inside the gearbox system of a WT to showcase the practicality of the proposed workflows. The contributions of this work aim to improve the reusability of the WT system models while ensuring data consistency and interdependencies between different system solutions, and to provide structured virtual testing workflows in the system architecture to support the virtual validation of the redesigned WT system easily. In addition, the work proposes an automated virtual testing workflow that can execute the appropriate models according to the specific test needs of the engineer to achieve the minimum necessary calculation work while obtaining sufficient results at the same time.

This work is organized as follows. Section 1 introduces the research background, the related works, and the blank spot in the state of the art in detail. Following, the studied WT as well as the models used for demonstration are described in Section 2. Section 3 describes the function-oriented system architecture and domain model classification of the WT system. Section 4 introduces the virtual testing workflows in SysML based on the function-oriented system architecture. Section 5 evaluates the superiority of the proposed workflow with a demonstration of virtual validation processes of partial WT components. Section 6 discusses the findings, the superiority, as well as remaining challenges of the proposed method and prospects for future research. Finally, Section 7 concludes the work.

## 2. Wind Turbine System and Multi-Domain Models

The research subject of this work is a WT system and the subsystems that compose it (see Figure 1). A WT is a device that uses the kinetic energy (wind energy) of airflow to generate electricity. The main component of the WT is the nacelle that converts the flow of mechanical energy into electrical energy. The nacelle also contains its components, such as the mechanical component gearbox that contains the HSS bearings. Virtual validation of the technical system (e.g., WT system) is related to the analysis of the behaviors of the system, e.g., lifetime, and load performance, which require the use of multiple domain models. This research focuses on the load performance analysis models of the nacelle, a failure analysis model and a smearing model of the bearing, which are introduced in detail in the following sections. During a virtual testing process, the nacelle model can be used to analyze the load performance of the bearing. As a sub-model of the nacelle system, the MBS model of the gearbox is composed of bearing MBS models with different fidelities, which lead to different accuracy of calculation results. Depending on different test needs, the calculation results can be used in different domain models, such as bearing failure and smearing models.

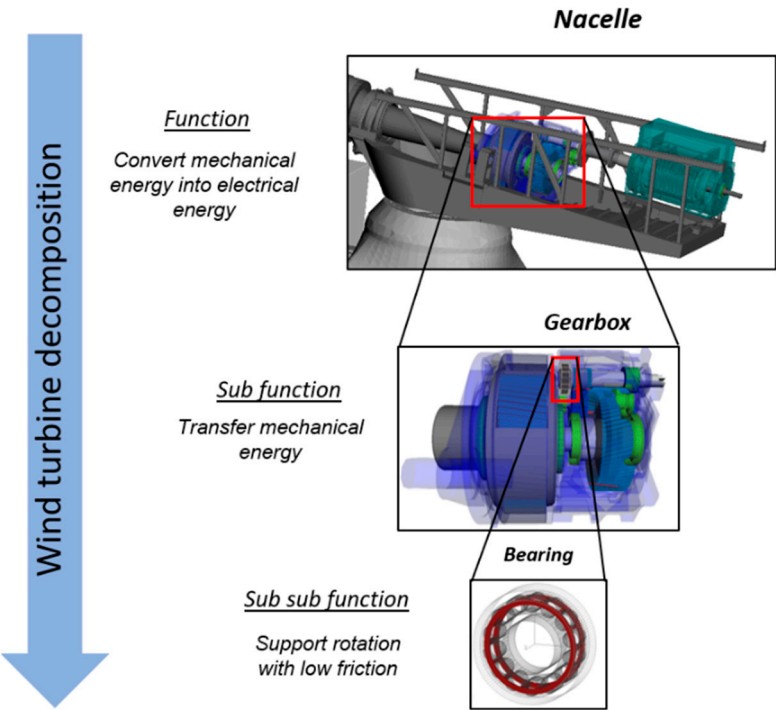

**Figure 1.** A wind turbine system with hierarchy level.

### 2.1. Load Performance Analysis Model

Figure 2 shows the MBS model used for bearing load performance analysis at two different fidelities. Choosing an appropriate modeling fidelity for a model is highly dependent on which stage of development the system is investigated. The modeling fidelity has a high impact on the runtime and also the number of outputs of the simulation. Therefore, choosing the right modeling fidelity for each test need can save a lot of runtime while still achieving the desired results. Two MBS models of bearings within a WT gearbox are used for two virtual testing needs in this research, respectively. The first test need is to perform a fatigue analysis according to ISO 281. The second test need is the evaluation of the risk of smearing.

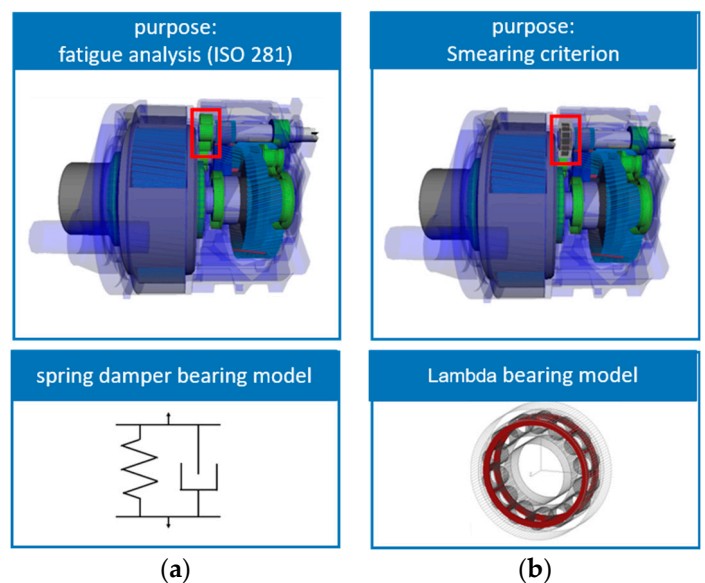

**Figure 2.** (**a**) A spring damper bearing MBS models within the WT gearbox; (**b**) A Lambda bearing MBS models within the WT gearbox.

## 2.2. Fatigue Analysis Model

The basic rating life $L_{10}$ as a criterion of bearing fatigue analysis has proved satisfactory for many years. A standardized formula, also known as the catalogue method [30], is the common means of calculation of a bearing's basic rating life. ISO 281 is applicable to the bearing under continuous rotation subjected to axial and radial loads. According to the ISO 281 guidelines, the basic rating life for a radial roller bearing is given by the equation:

$$L_{10} = \left(\frac{C_r}{P_r}\right)^{10/3}, \qquad [hours] \qquad (1)$$

where $C_r$ denotes the basic dynamic radial load rating for a radial roller bearing which is calculated as:

$$C_r = b_m f_c (i L_{we} cos\alpha)^{7/9} Z^{3/4} D_{we}^{29/27}, \qquad [N] \qquad (2)$$

$C_r$ is based on geometrical and manufacturing properties of bearing such as the row count $i$; the roller diameter, $L_{we}$; the number of rollers $Z$; the roller diameter, $D_{we}$; and the contact angle, $\alpha$. The factor $b_m$ equals 1.1 for the cylindrical roller bearings, whereas the factor $f_c$ depends on the geometry of the bearing and has been chosen according to ISO 281:2007. $P_r$ refers to the dynamic equivalent load, which is equal to bearing radial load $F_r$ for radial roller bearing subjected to radial load only with $\alpha = 0°$ [30]. As all these parameters are known at the manufacturing stage, the load rating will be usually generated after the bearing designation is determined.

For this test need, it is not necessary for the model to explicitly represent all rollers and the cage. The kinematics of the rollers are not needed as an input of ISO 281. Therefore, a spring damper MBS model using the characteristics of the bearing is sufficient (see Figure 2a).

## 2.3. Smearing Analysis Model

Smearing is a dynamic driven damage pattern in roller bearings. Smearing damage risk can be calculated using state of the art criteria [31–33]. Fundamental to these equations is the friction power intensity [31]. Dyson concluded that the occurrence of smearing during tribo-testing is always accompanied by an increase in measured friction. Dyson also stated that smearing can only be generated when significant slip is present between the contact partners. In this work, the criterion defined by van Lier is used. It is based on the power P per loaded area A and depends on the friction coefficient $\mu$, the difference in circumferential velocity of the contact partners $u_1(t) - u_2(t)$, and the maximum Hertzian contact pressure $p_{max}(t)$:

$$\left(\frac{P}{A}\right)_{max} = max(0.5 \cdot \mu \cdot p_{max}(t) \cdot (u_1(t) - u_2(t))), \qquad [W/mm2] \qquad (3)$$

In order to evaluate the risk of smearing, the loading as well as the kinematics of each roller of the bearing have to be calculated. Therefore, a detailed MBS model (see Figure 2b) is implemented on the rotor side of the HSS [34]. This model explicitly represents the outer ring, inner ring, cage, and rollers as well as the lubricant and its film thickness in the contact zone. A more detailed bearing model delivering the necessary outputs also increases the runtime significantly due to its large amount of degrees of freedom.

## 3. System Modeling Approach

This section demonstrates the system modeling approach in detail through the WT system case study. As described in Section 1, the system model is a centric model in the modeling process that can help manage risk and complexity by capturing comprehensive information, which can be graphically presented to system engineers in different views as needed. The system model adopts the system architecture to provide the necessary structure, logic, syntax, and semantics to define system elements (e.g., requirements, system

functions, physical solutions), so to achieve a consistent understanding of the system. In order to increase the traceability between system elements, a function-oriented system architecture proposed by [26,35] is applied to the WT system modeling in this work.

### 3.1. Function-Oriented System Architecture

In the function-oriented system architecture, functions can be derived from functional requirements. As shown in Figure 3, the WT system function is decomposed into sub-functions, which serve as the parts of one higher-level function. A system function, or a part of it, is delimited by a boundary. Through defined interfaces, physical quantities can enter and leave the function as functional flows. These flows can be energy flows, material flows, or signal flows. The delimited function transforms the quantities of the incoming flows to other quantities of the outgoing flows. Functions are referred to as elementary functions if the transformation of the flows they represent does not decompose further [36]. A solution will inherit the functional flows from the function it fulfills through the generalization relationship. The solution consists of the physical effects and further geometric elements such as surface pairs, between which a physical effect occurs. Physical effects and geometric elements are defined at the parametric level. In addition, solutions are composed of sub-solutions with a hierarchical structure, so that the parameters of the solution can be shared across the hierarchy. However, using physical formulas and effect surfaces is only the conceptual design and cannot meet the simulation accuracy required to solve engineering problems. Therefore, the domain models that evolve from the evolution of the concept designs are integrated into the system model as a part of the solution element for further description of the system. Each domain model is encapsulated in a constraint in SysML with input and output ports, which are integrated with the external domain simulation models. These input and output ports are connected to the parameters from the solution elements of the system architecture [26].

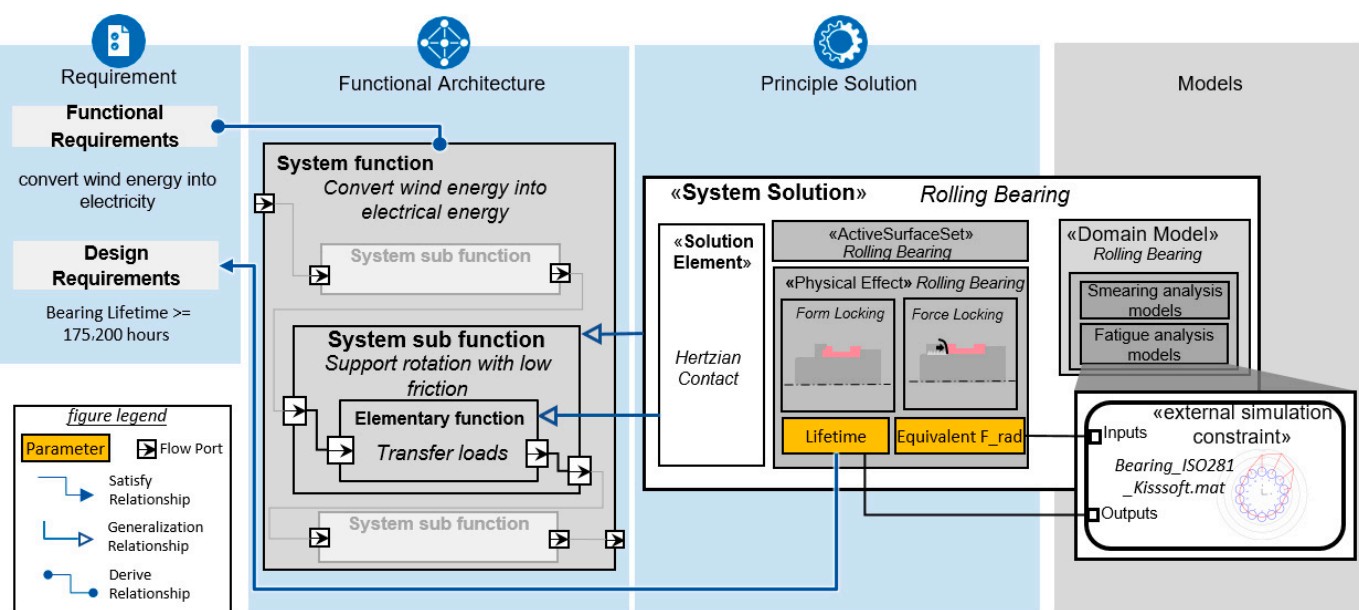

**Figure 3.** The schematic of function, solution, and domain model architecture in a system model.

### 3.2. Domain Model Classification

The domain models introduced before are represented in different external engineering software tools with different precision and different calculation complexities. Domain models further describe a solution in different scopes, analyze the solution with different purposes, and have different detail levels and parameter requests. A complex system requires a large number of domain models to represent. To improve the reusability of domain models, it is necessary to classify these domain models. This work represented

these domain models in a three-dimensional model matrix according to the scope, purpose, and fidelity of the models [26].

As described in Figure 4, first of all, each domain model should have a clear system scope. There is a hierarchical structure between the different scopes of the system that corresponds to the solution hierarchy in the system structure. A large system scope (e.g., nacelle system) is composed of small scopes of the system (e.g., bearings, gears, generator systems). Secondly, each model should have a clear purpose which refers to the analysis goal of the developer. This can be the evaluation of the efficiency of the system (e.g., annual energy production model), performance measures (e.g., MBS load calculation model), or failure mode test models (e.g., fatigue or smearing analysis model). At last, in order to ensure the reliability of the test results, the model requires a certain degree of accuracy and fidelity. The domain models with the same purpose may have different fidelities. For example, a bearing can be modeled regarding the NVH performance in MBS as a spring damper model, or as a detailed bearing model also considering the kinematics of each roller of the bearing. Domain models can be developed in a modular way (e.g., MBS system and subsystem models), where the modeling defined in the sub-model is available in its parent models. Sub-models can also contain substructures allowing substructures to be nested. This provides great flexibility and modular structure to the domain model. Modular construction means different levels of detail and fidelity can be easily accomplished in different model variants. Therefore, the fidelity of a large scope may be determined by the combination of fidelities of the small scopes. The engineer can select the required fidelity for a small system scope and combine them into a large system scope. For example, the MBS model of a nacelle system can be composed of HSS bearing models with two different fidelities, which also leads to two different fidelities of the MBS model of the nacelle system. The model matrix defines a standardized way of model classification. This improves the reusability of the domain models and allows engineers to quickly add and access required domain models. The more models are organized in the model matrix, the greater the benefit of the model classification.

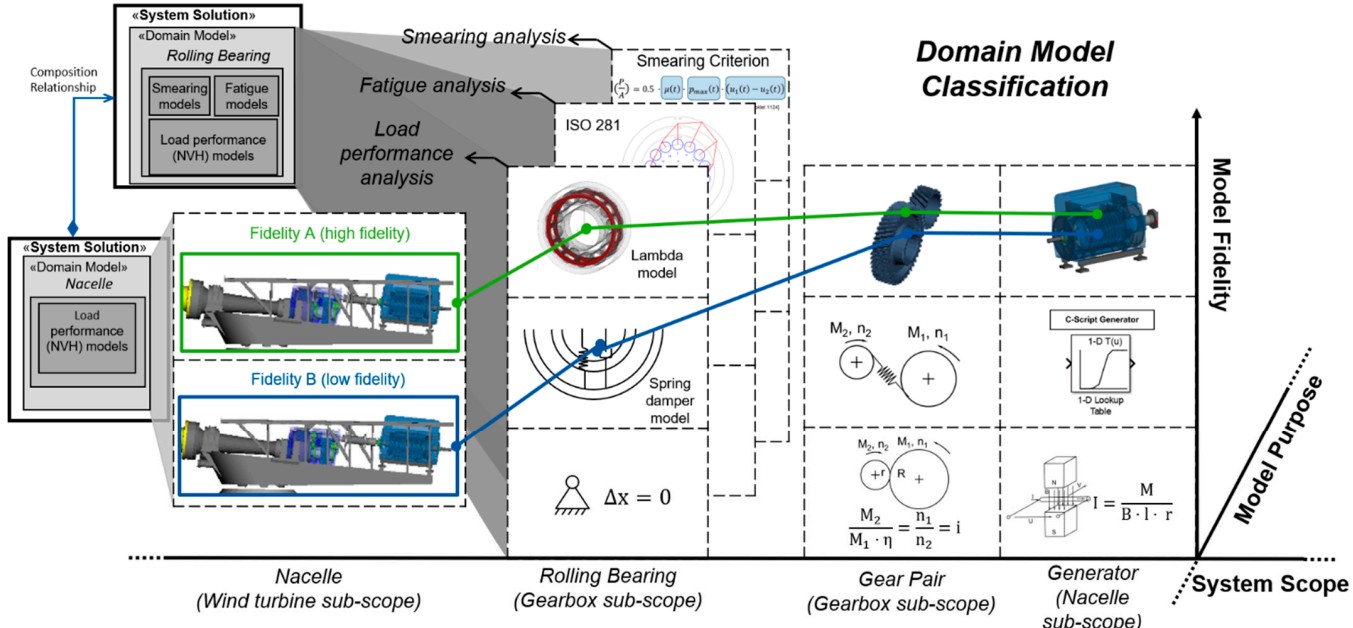

**Figure 4.** The classification of domain models based on system scope, model purpose, and model fidelity.

## 4. Design of Virtual Testing Workflows in SysML

As described in Section 3.1, the function-oriented architecture starts by deriving system functions from functional requirements or use cases. These functions are implemented by solution elements. To virtually validate that a function is satisfied by the corresponding solution, multiple virtual testing workflows should be designed regarding different test needs, such as smearing analysis testing and fatigue analysis testing. The virtual testing workflows model and perform a series of simulation activities. In order to perform specific simulation activities, engineers usually need to select appropriate domain models and manually sequence the simulations of these domain models for creating virtual testing workflows. Therefore, to reduce manual efforts for sequencing the workflows, it is necessary to establish a general and standardized design method for multiple virtual testing workflows, which helps engineers to easily design and reuse the workflows with different test needs.

### 4.1. Nested Structure of Virtual Testing Workflows

The virtual testing workflows adopt a modular design [37] that subdivides the workflows into smaller parts called modules. The simulation activity is regarded as a module for performing the corresponding simulations of the domain models. Modular simulation activities provide great flexibility in the design of virtual testing workflows. In addition, as mentioned in Section 3.2, domain models are classified according to system scope, model purpose, and model fidelity. The simulation activities are nested based on the classification of the domain model. For example, a simulation activity regarding a specific model purpose includes sub-simulation activities that execute the models with different model fidelities.

As shown in Figure 5, the proposed workflow employs a nested design framework, in which the system scopes should be considered at the top level in the nested structure. It ensures that effort and resources are not wasted on tasks that do not contribute to the current analysis work (i.e., validation of the solutions not in focus). Before the workflows start, engineers manually set a test need, such as *"Smearing testing"* and *"Fatigue testing"*. An activity called *"Check test needs"* is created at the *"System scope"* level of the workflow to capture test needs from engineers. A decision node needs to be created after the activity *"Check test needs"*. In this work, the symbol of the decision node is a diamond that connects with one object flow as the input and two or more control flows as the output. The test needs are passed by the activity *"Check test needs"* to the decision node through the object flow. The control flows model a transition between the decision node and virtual testing activities (i.e., *"Fatigue testing"* and *"Smearing testing"*). Control flows can define the guard condition. A guard condition is a Boolean condition that is evaluated when a transition is initiated. The Boolean expression is contained in the square bracket to evaluate the input (i.e., test needs) of the decision node. The transitions between the decision node and virtual testing activities only occur when the guard condition is evaluated to be true.

Each virtual test involves various related domain models from different system scopes with a specific purpose. Domain models for a specific purpose require parameters as input that may be determined in models for other purposes. Therefore, at the *"Model purpose"* level of the workflow, engineers should also consider the simulation sequence between each domain model. Once the sequence is established, the virtual testing workflow can be reused. For example, when conducting *"Smearing testing"*, the workflow always performs the *"Load performance models"* at first and then performs *"Smearing analysis models"* in sequence.

Furthermore, each domain model has a certain degree of fidelity. Therefore, the *"Model purpose"* level of the workflow needs to be further designed to determine the fidelity of the domain models in a workflow. Engineers can either directly execute workflows based on pre-set model fidelity or set the appropriate model fidelity before the workflow begins. Similar to *"System scope"* level design, the appropriate model fidelity is determined at the decision node while the workflow is running. For example, at the *"Model fidelity"* level of the workflow, the workflow executes a high-fidelity load performance model (e.g., *"Lambda model"*) to support motion analysis for smearing testing.

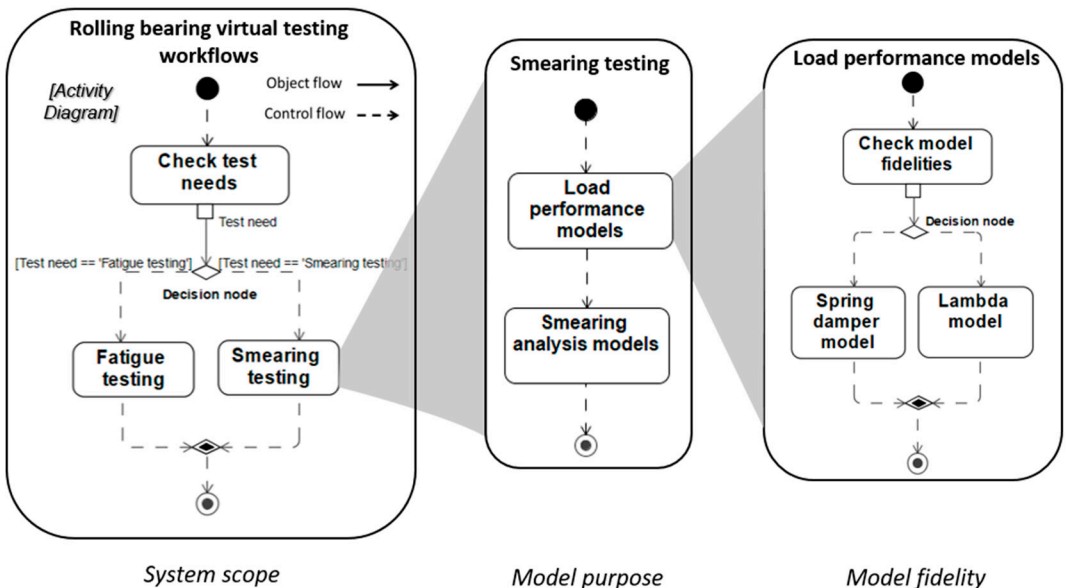

**Figure 5.** Nested virtual testing workflows based on the domain model classification.

### 4.2. Virtual Testing Workflows for Handling Simulation Results

When the appropriate fidelity model is selected, the workflow will perform the corresponding simulation of the domain model. System data is continuously updated as the domain models are simulated. Engineers usually manually update the data based on the simulation results and used the data in the next simulation. Therefore, to reduce these manual efforts and improve the automation of the workflow, it is necessary to further design the workflow for handling simulation results, which support the up-to-date data during the running of the workflow.

As shown in Figure 6, the simulation activities are used to trigger simulations of related domain models, which are executed in external modeling tools. The constraint block in SysML encapsulates the domain models as an expression with inputs and outputs, linking to external tools via programming languages such as MATLAB. The system parameters can be connected with each domain model through the ports of the constraint block, thus realizing the data synchronization with the external simulation models.

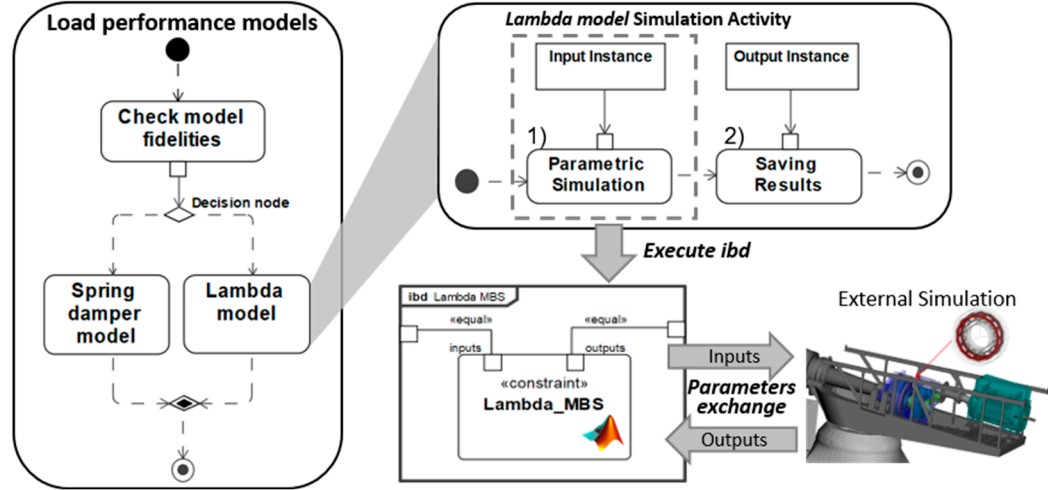

**Figure 6.** Handling simulation results to support up-to-date data during virtual testing.

To keep the data up-to-date during testing, each simulation activity includes the following two actions:

- Action 1 (*"Parametric Simulation"*): Obtaining the latest updated data and executing simulations based on those data;
- Action 2 (*"Saving Results"*): Saving the simulation results and updating the data for the subsequent simulations or requirement verifications.

In this work, the system data are saved in the instances of SysML. In SysML, an instance is a specific occurrence of any object. The instance specifies an object (e.g., a domain model) in SysML with specific property values. In this work, these instances provide data for the system model and parameterize the solutions and domain models in the system model. These instances are used in the action *"Parametric Simulation"* and *"Saving Results"* as the target object to be executed and the target object of where to save the results. The instance can be passed as an argument to the actions as a whole object. By using the API scripting language of CSM, the action *"Parametric Simulation"* will invoke the given objects and execute the parametric simulation, while the action *Saving Results* will wait until the simulation is completed and save the simulation results to the given objects.

Take the *"Lambda model"* simulation activity as an example. When the simulation starts, a runtime object (i.e., *"Input Instance"*), wherein the input parameters are stored, will be passed as an argument to the action *Parametric Simulation*. Then, the action will execute the constraint blocks called *"Lambda_MBS"* in the internal block diagram (ibd) in SysML to perform the corresponding external simulation (i.e., *"MBS model"*) based on the input instance. Parameters in the system model are exchanged with external models through constraint blocks in the ibd. After executing the simulation of the domain model, the simulation results are returned to the runtime object of the system model through the constraint block interface. A runtime object (i.e., *"Output Instance"*), wherein the output parameters are temporarily stored, will be passed as an argument to the action *"Saving Results"*. The action *"Saving Results"* saves the outputs from the runtime object in a given instance, in which the saved values will be used for other simulations or the requirement verifications.

The following advantages are observed using instances in the system model. The instance can be used to define specific configurations of system solutions, including the values of their properties (e.g., parameters), and is therefore potentially executable and can be simulated. Multiple instances with different values of properties can be created for a system solution, which greatly improves the reusability of the system model. There is no dependence between all kinds of instances, therefore, engineers are able to deal with parallelization of the simulations between different solution instances.

*4.3. Virtual Testing Workflows in the Function-Oriented Architecture*

This work combines proposed virtual testing workflows with the function-oriented system architecture to ensure that different engineers can conduct specific virtual tests under the premise of using consistent data. The feasibility of a simple virtual testing workflow in a function-oriented system architecture has been demonstrated in [29].

As shown in Figure 7, the proposed virtual testing workflow consists of a series of pre-ordered activities for performing simulations regarding the domain models of the WT system in sequence. These domain models are distributed at different hierarchy levels of the system. The workflows (e.g., smearing and fatigue calculation of rolling bearings) can be defined in the function-oriented system architecture as the stereotype *«workflow»* and represented in the activity diagrams of SysML. The function-oriented system model provides a set of diagrams (e.g., internal block diagrams) to represent system information and show the interaction between different system elements and connections between different parameters. For example, the equivalent radial force (*"Equivalent F_rad"*) is one of the parameters of the physical domain. This parameter can be bound to constraint blocks in different domain models with a specific purpose and fidelity (e.g., *"ISO 281 model"* and *"Smearing Criterion model"*) through binding connectors. The binding connector specifies

that properties at both ends of the connector have equal types and values, which ensures that the parameters used in the different domain models come from the same source. Same as described in Section 4.2, the parameter values can be specified in the instances, and the constraint blocks can be triggered when workflows run.

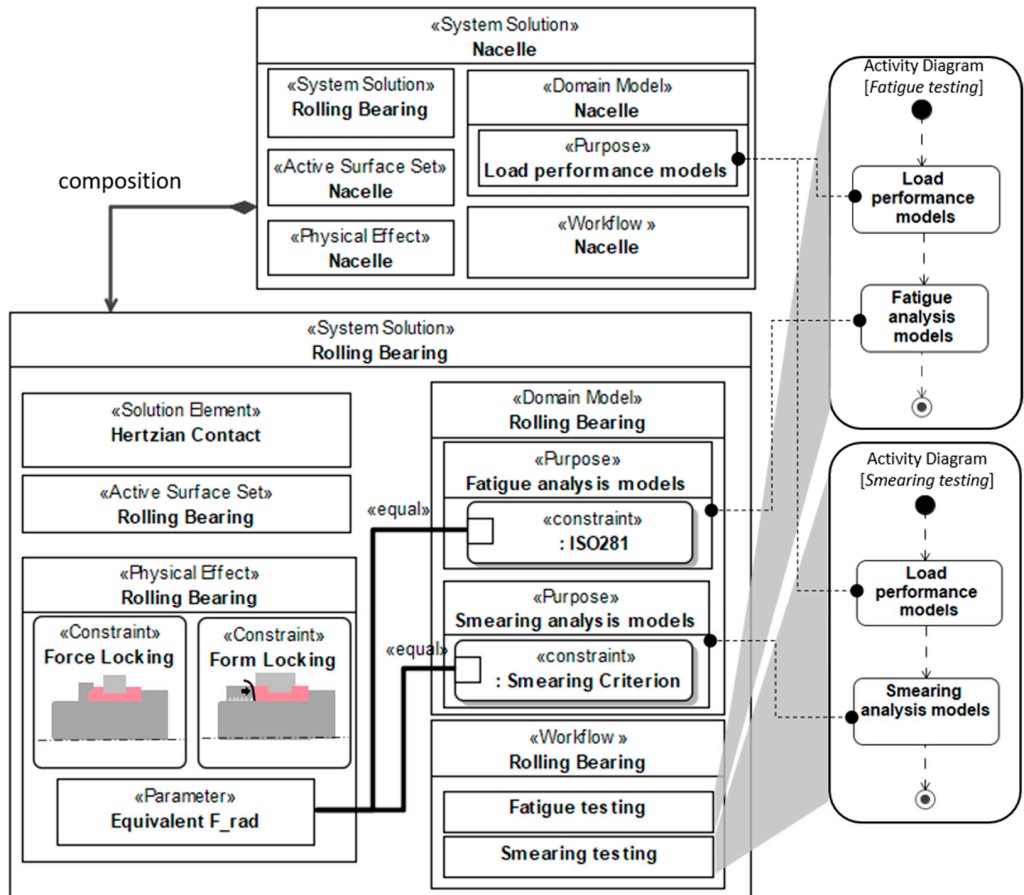

**Figure 7.** Virtual testing workflow in the function-oriented system architecture.

In conclusion, compared to the traditional workflows stored in documents, the model-based workflows in SysML activity diagrams avoid the independent development and subsequent integration of each discipline. This greatly improves the efficiency while ensuring the accuracy of the system validation process, thereby reducing the risk of failure in the late development stage. In addition, the work proposes a design method for a standardized virtual testing workflow framework based on domain model classification, which improves the efficiency of the virtual testing process. Next, an executable workflow is achieved at the parameter level based on a system architecture in SysML. Virtual testing workflows make the data in the system model alive, create continuous process links between the partial domain models that enable data consistency during the virtual testing processes, and allow change propagation.

## 5. Case Studies and Results

The proposed virtual testing workflows support system virtual validation after the redesign of the WT system. A virtual validation of the bearing system within a WT is presented in this section to show the usage of the workflow. The virtual validation of the bearing system provides the testers with confidence that the solution (i.e., bearing system) satisfies the function by considering the fulfilment of the related requirements. In this case study, there are two design requirements which need to be validated: the bearing's lifetime related to fatigue analysis, and the maximal criterion value defined by van Lier [33]

to evaluate the risk of smearing in the bearing system. The simulation results (e.g., the lifetime) are calculated and saved as value properties of the solution, which are connected to the non-functional requirement through the satisfy relation.

### 5.1. System Model of System Bearing

As described in Section 3.1, the WT system model applies the function-oriented system architecture. For example, the functional architecture provides the structure for the solution "*Rolling Bearing*" of WT with its describing parameters (e.g., the mean rotation speed). In order to solve real-world engineering problems, the bearing system is further represented by the domain models, which are load performance analysis models with two different fidelities, fatigue analysis model, and smearing analysis model. The fatigue analysis model is created in the domain-specific software KISSsoft [38] to perform failure analysis based on ISO 281. The smearing analysis model is created in MATLAB software to perform the smearing risk analysis.

The fatigue analysis model is taken as an example (see Figure 8). The parameters of the "*Rolling Bearing*" can be connected to the standardized interface ports on the constraint block in the internal block diagram of SysML. Constraint blocks contain MATLAB scripts that can trigger external models, so as to integrate the external simulation into the system model. The fatigue analysis model refers a partial of parameters from the "*Rolling Bearing*" solution. The mean value of rotation speed "*MeanRotationSpeed*" and equivalent radial load acting on the bearing "*Equivalent F_rad*" are regarded as the input parameters of the "*Bearing_ISO281_Kisssoft*" constraint and can be transferred to the external fatigue analysis model through the MATLAB script file "*Bearing_ISO281_Kisssoft.m*" contained in the constraint block. After the calculation is completed, the key output parameters (i.e., "*Lifetime*") will also be returned to the system model as the outputs through the interface ports in the constraint block. The output results are used for subsequent simulations or requirement validations.

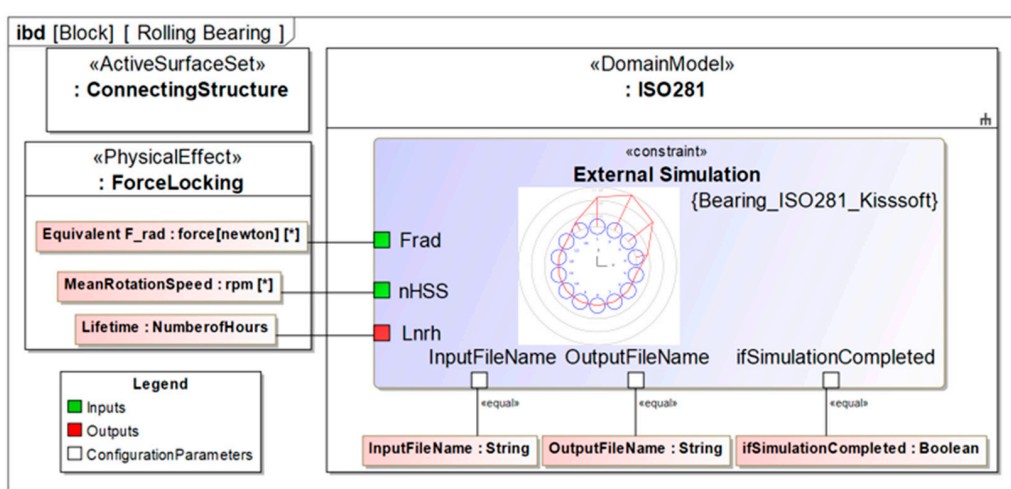

**Figure 8.** The fatigue analysis of a WT bearing system in the internal block diagram.

### 5.2. Virtual Testing of Bearing System

For the demonstration of the workflows, the model of the WT research nacelle of the "Forschungsvereinigung Antriebestechnik" (FVA) is used (see Figure 9). The research nacelle consists of the main elements of the main shaft assembly system, gearbox, high-speed asynchronous generator, and full power converter system.

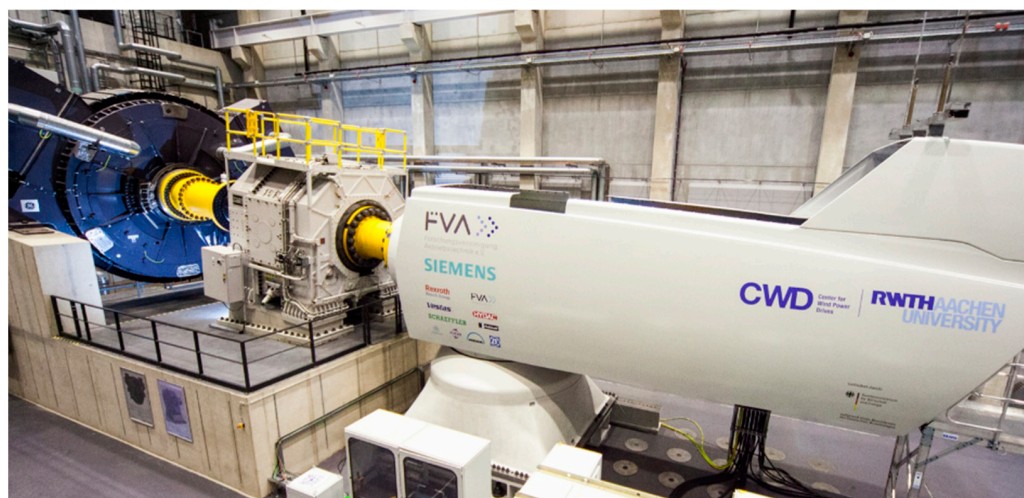

**Figure 9.** The FVA WT nacelle on test bench.

The studied gearbox consists of one planetary stage and two helical gear stages and has a gear ratio of around 63. The gear pairs of the gearbox are modeled using a gear pair force element to simulate the behavior of gear pairs. The generator is modeled as a three-phase asynchronous induction generator, and the specific parameters of the generator are provided by the manufacturer (see Table 1). A cylindrical roller bearing "NU 2338" is selected as the HSS bearing of the gearbox, and the geometric parameters of the bearing are from the FVA database (see Table 1). The model specifics and detailed descriptions are given in [39].

**Table 1.** Parameters of the generator and the bearing.

|  | Parameter | Value |
|---|---|---|
| **Generator** | Power | 3 MW |
|  | Rated Voltage | 720 V |
|  | Rated Current | 2564 A |
|  | Rated Torque | 24.7 kNm |
|  | Rated Speed of rot. HSS | 1100 rpm |
| **Bearing** | Inner Diameter | 190 mm |
|  | Outer Diameter | 400 mm |
|  | Width | 132 mm |

As mentioned in Section 2, the gearbox MBS model validated by IEC 61400 is linked to the analytical models for the WT and generator-converter control in order to simulate realistic WT behavior. To quantify the risk of gearbox bearing damage due to electrical faults, this work presents two virtual testing scenarios. For the first usage scenario, the need of the test is the failure analysis of the bearing. For the second usage scenario, the test need is bearing wear analysis.

In the first usage scenario, the bearing lifetime requirement is set to be greater than 20 years (i.e., 175,200 h) [30]. The engineer needs to validate that the system design satisfies this requirement through a fatigue analysis workflow. The following Figure 10a shows the calculation results and calculation time of the validations using domain models with two different fidelities. The calculation results show the small differences in the load, speed, and lifetime parameters, while both tests meet the lifetime requirements well. However, choosing a spring-damper (low-fidelity) model has a more obvious advantage in calculation time, that reduces calculation time from 3486 to 137 s.

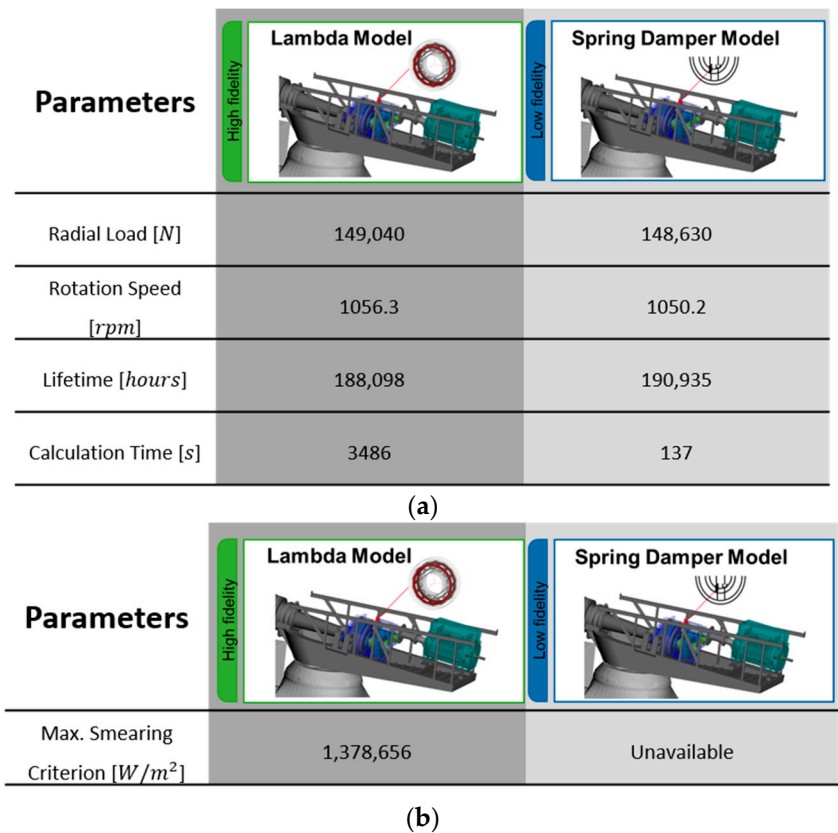

**Figure 10.** (**a**) Comparison of calculation results of using two fidelity MBS domain models in scenario 1; (**b**) Comparison of calculation results of using two fidelity MBS domain models in scenario 2.

In the second use scenario, the loads acting on the HSS of the gearbox have changed due to the short circuit fault. In this case, engineers need to validate the system again. The change caused by the generator significantly increases the failure risk of the bearing in regard to the smearing. Therefore, smearing analysis should also be considered in the more comprehensive validation process of the bearing system. However, this requires high-fidelity load calculation to obtain bearing kinematics. Figure 10b shows that the smearing criterion data of the bearing system under the special load case can only be calculated by the high-fidelity model, but is not available in the low-fidelity model. The calculated smearing criterion is 1,378,656 $W/m^2$ , which does not meet the smearing requirement. In this case study, the maximal smearing criterion should be less than the threshold, which is set as 1,000,000 $W/m^2$ [33]. Therefore, the system is virtually diagnosed as faulty under the special load case.

## 6. Discussion

Case studies demonstrate that, for complex technical systems, the proposed virtual testing workflows have advantages over classical virtual testing processes. First of all, the virtual testing workflows based on the function-oriented system architecture are executable. It not only manages the sequence of simulations, but also interacts with physical behavior models, offering the possibility to easily detect failure risks through high-precision simulations. Secondly, the case studies show that changes (i.e., load case changes) can be propagated across the various domain simulation models through the central system model to ensure data consistency. When load case changes, the risk of bearing smearing failure can be detected in time. Therefore, using the workflows can prognose the failure risks of technical systems due to redesign between different domain simulation models. At last, a structured workflow is necessary because when systems become complex, it will

improve the reusability of the virtual testing workflows and the efficiency of the validation process. These case studies demonstrate that this workflow ensures that testers can easily perform the appropriate virtual testing scenarios, thereby avoiding precision problems or waste of computing resources. While this work demonstrates how to select the appropriate workflow based on the testing needs, it should be noted that the proposed workflows also support adding more virtual testing processes to ensure that a solution satisfies the corresponding function.

However, the current workflow still has limitations. First of all, the introduction of newly proposed workflows in the industry requires broad understanding and acceptance within and across disciplines. Therefore, user-friendly operation interfaces and standards for data exchange between domain models need to be created. Secondly, a function always contains several sub-functions that need to be validated, and the corresponding solutions are also distributed in hierarchy levels. Therefore, it is necessary to further standardize workflows to implement virtual validation across all levels of the functional hierarchy. In addition, this virtual verification uses a pre-set simulation sequence. Once the solution is reused in a different system, the workflows have to be manually restructured and sequenced based on the system context to ensure that the tests can still be carried out correctly. This issue can be improved by modularizing the simulation activities in future work. At last, in this work, engineers need to check the validation results after one design iteration loop and decide whether the next iteration can start or not until the design converges toward a solution that fulfills the requirement. Given that these design loops can be implemented through optimization algorithms or cost functions, the degree of automation of the proposed workflow can be further increased.

## 7. Conclusions

In order to eliminate the barriers caused by the inconsistency of document definitions between multiple disciplines of a system during the development process, this research proposes an executable validation workflow that can be integrated with external domain simulation models based on a function-oriented system architecture model. The parameter values in the system model are used in the external domain simulation models and then updated and reused after the simulation runs to ensure that the data is up-to-date and changes are propagated to the various domain models. This work constructs a test process based on model purpose, system scope, and model fidelity according to domain simulation models. The system engineers can perform virtual validation of the required system by using the workflow. These concepts are demonstrated through a wind turbine system. In the use case, the bearing of the wind turbine system is tested under different load conditions with minimal necessary computational efforts while obtaining sufficient results. The testers can easily use the workflow to meet different test needs. The contribution of this research is to increase the reusability of models based on a clear classification scheme while ensuring a data-consistent development process. More virtual testing processes can be introduced into this standardized workflow, thus improving the generalization of the validation process. The proposed workflow also can detect the risk of system failures in the early stages, thus reducing development costs. This workflow needs to be further standardized in future work to address the challenge of functional verification across the hierarchy levels. At the same time, in order to support the reuse of the solution, it is necessary to consider how the virtual testing workflow can be quickly rebuilt in different systems.

**Author Contributions:** Conceptualization, Y.Z., J.R., G.J., J.B. and G.H.; methodology, Y.Z., J.R., G.J., J.B. and G.H.; software, Y.Z., J.R.; validation, Y.Z.; formal analysis, Y.Z., G.H., J.B.; writing—original draft preparation, Y.Z., J.R.; writing—review and editing, Y.Z., J.R., G.J., J.B. and G.H.; supervision, G.J., J.B. All authors have read and agreed to the published version of the manuscript.

**Funding:** This research received no external funding.

**Institutional Review Board Statement:** Not applicable.

**Informed Consent Statement:** Not applicable.

**Data Availability Statement:** Not applicable.

**Conflicts of Interest:** The authors declare no conflict of interest.

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
