# Peer review of "Virtual Testing Workflows Based on the Function-Oriented System Architecture in SysML: A Case Study in Wind Turbine Systems"

_2674-032X, doi:10.3390/wind2030032_

Round 1
Reviewer 1 Report
the work is well written and well presented, it could be published after addressing minor changes (please refer to the attached file).

Author Response
Please see the attachment, thank you!

Reviewer 2 Report
In this paper, the authors carried out the study regarding virtual testing workflows based on the function-oriented system architecture for wind turbine systems in SysML. The research topic is meaningful. The main problems in this paper are as follows:
Comments:
(1) It is suggested that the content of Section 2 be merged into Section 1 "Introduction".
(2) Line 283, 354, 467 and 508: “Error! Reference source not found.” shoud be removed.
(3) In Section 6, the basic information and parameters of the two Cases should be briefly introduced.
(4) Please arrange the format of references according to the requirements of the journal.
Author Response
Please see the attachment, thank you!

Reviewer 3 Report
The manuscript "Virtual testing workflows based on the function-oriented system architecture for wind turbine systems in SysML" focuses on the discussion of the predictive virtual model wind turbine concerning bearings workflow.
The introduction and the second section are devoted to reviewing the wind turbine virtual testing approach area. The Authors are discussing Model-based Systems Engineering and Virtual Validation of Technical Systems approaches. The sections have a strong structure and reasonable conclusions.
The third section сontains the correct mathematical description of the wind turbine bearing models (by using spring damper and lambda models) and its Fatigue Analysis Model Smearing Analysis Model.
Fourth and fifth section deal with the description of wind turbine architecture in the Systems Modeling Language. The description of the workflows with regard to wind turbine bearings is done.
The sixth section focuses on virtual testing of bearing system results. I suggest including in the description information about hardware explanation.
In the seventh section the Authors discussion about future research.
In the Conclusion section, the Authors sum up the results of their work.
This paper is well written, but it seems like a case study, so I suggest renaming the manuscript. Also, the Authors should add the references 2020-2022s in the literature review.
Author Response
Please see the attachment, thank you!
